# Novel Cyclophilin Inhibitor Decreases Cell Proliferation and Tumor Growth in Models of Hepatocellular Carcinoma

**DOI:** 10.3390/cancers13123041

**Published:** 2021-06-18

**Authors:** Sonia Simón Serrano, Michele Tavecchio, Alvar Grönberg, Wondossen Sime, Mohamed Jemaà, Steven Moss, Matthew Alan Gregory, Philippe Gallay, Eskil Elmér, Magnus Joakim Hansson, Ramin Massoumi

**Affiliations:** 1Translational Cancer Research, Department of Laboratory Medicine, Lund University, Medicon Village, 223 63 Lund, Sweden; sonia.simon_serrano@med.lu.se (S.S.S.); wondossen.sime@med.lu.se (W.S.); jemaamohamed@gmail.com (M.J.); 2Abliva AB, Medicon Village, Scheelevägen 2, SE-233 81 Lund, Sweden; micheletavecchio@libero.it (M.T.); alvar.gronberg@abliva.com (A.G.); eskil.elmer@med.lu.se (E.E.); magnus.hansson@abliva.com (M.J.H.); 3Mitochondrial Medicine, Department of Clinical Sciences, Lund University, BMC A13, SE-221 84 Lund, Sweden; 4Isomerase Therapeutics Ltd., Suite 9, Science Village, Chesterford Research Park, Cambridge CB10 1XL, UK; steven.moss@isomerase.co.uk (S.M.); matt.gregory@isomerase.co.uk (M.A.G.); 5Department of Immunology & Microbial Science, The Scripps Research Institute, La Jolla, CA 92037, USA; gallay@scripps.edu

**Keywords:** PPIase, cyclophilin, hepatocellular carcinoma, cell cycle and mitosis

## Abstract

**Simple Summary:**

Cyclophilins, a family of proteins with peptidyl prolyl isomerase activity, have been found to be overexpressed in several cancers, including hepatocellular carcinoma (HCC), and their expression is correlated to a poor prognosis. Cyclophilins play an important role in proliferation and cancer resistance in HCC. In this study, we evaluated the potential capacity of cyclophilin inhibitors as a treatment against HCC. We showed that our selected cyclophilin inhibitor, NV651, was able to decrease cell proliferation in vitro and induce an accumulation of cells in the G_2_/M phase due to a mitotic block. We could also confirm its capacity to decrease tumor growths in mice and its safety in vitro as well as in vivo.

**Abstract:**

Hepatocellular carcinoma (HCC), the most common primary liver cancer, is usually diagnosed in its late state. Tyrosine kinase inhibitors such as sorafenib and regorafenib are one of the few treatment options approved for advanced HCC and only prolong the patient’s life expectancy by a few months. Therefore, there is a need for novel effective treatments. Cyclophilins are intracellular proteins that catalyze the cis/trans isomerization of peptide bonds at proline residues. Cyclophilins are known to be overexpressed in HCC, affecting therapy resistance and cell proliferation. In the present study, we explored the potential of cyclophilin inhibitors as new therapeutic options for HCC in vitro and in vivo. Our results showed that the novel cyclophilin inhibitor, NV651, was able to significantly decrease proliferation in a diverse set of HCC cell lines. The exposure of HCC cells to NV651 caused an accumulation of cells during mitosis and consequent accumulation in the G_2_/M phase of the cell cycle. NV651 reduced tumor growth in vivo using an HCC xenograft model without affecting the body weights of the animals. The safety aspects of NV651 were also confirmed in primary human hepatocytes without any cytotoxic effects. Based on the results obtained in this study, we propose NV651 as a potential treatment strategy for HCC.

## 1. Introduction

In 2018, liver cancer, with an incidence of 841,080 new cases and a toll of 781,631 deaths, was classified as the sixth-most frequent cancer and the fourth-most common type of cancer-related deaths worldwide [1]. Hepatocellular carcinoma (HCC), the most common form of liver cancer, can appear as a consequence of viral infections such as Hepatitis B or C virus, aflatoxin exposure, chronic alcohol consumption and nonalcoholic fatty liver disease (NAFLD) [2]. HCC staging follows the Barcelona Clinic Liver Cancer (BCLC) system [3], which guides the treatment regimen. According to BCLC, when the tumor is detected at an early stage (stage 0), patients receive a curative treatment such as resection, transplantation or radiofrequency ablation (RFA), followed by trans-arterial catheter embolization (TACE) and sorafenib (stages A–C) and symptomatic treatment (stage D) [4]. Sorafenib can prolong the life expectancy by a few months [5] before resistance develops [6]. Recently, other tyrosine kinase inhibitors have been accepted for first-line systemic therapy, including Lenvatinib [7], and for second-line systemic therapy: Regorafenib [8], Cabozantinib [9] and Ramucirumab [10]. On the other hand, none of these novel compounds have shown a significant increase in the life expectancy in comparison to sorafenib [7,8,9,10]. Therefore, new drugs with novel mechanisms of action are urgently needed.

Cyclophilins are one of three families of peptidyl prolyl isomerase (PPIase) proteins (together with parvulins and FK506-binding proteins (FKBPs)) [11], and their role is to catalyze the cis/trans isomerization of peptide bonds at proline residues [12]. Several cyclophilins have been discovered in the human genome, the most studied being A, B and D, residing mainly in the cytosol, ER and mitochondria, respectively [13,14,15]. The most well-known cyclophilin inhibitor is Cyclosporin A (CsA), a non-ribosomal cyclic undecapeptide firstly isolated from the fungus *Tolypocladium inflatum* by Sandoz (now Novartis) [16]. It is a widely used immunosuppressant, due to its capacity to decrease the activity and proliferation of T-lymphocytes by the inhibition of calcineurin in a ternary complex formed with cyclophilin A (CypA) [17].

Cyclophilins have been indicated as potential drug targets for cancer treatment due to their overexpression in a variety of cancer types [18,19] and their involvement in several cancer cell activities, such as protein folding, proliferation and cell cycle regulation [20,21,22,23]. Cyclophilins have also been presented as potent antioxidants protecting cancer cells against elevated ROS levels, decreasing hypoxia-related death [24,25]. Moreover, cyclophilins were found to increase the expression of ATP-binding cassette (ABC) transporters, reducing intracellular drug accumulation and potentiating therapy resistance [26]. Chemical inhibition with CsA or Sanglifehrin A (SfA) caused a synergistic effect with cisplatin, a DNA-damaging reagent, by increasing the cell death in HCC and ovarian cisplatin-resistant cancer cells due to the antioxidant activity and potential decreased expression of genes involved in DNA damage repair [27,28].

In the present study, we designed and evaluated a new cyclophilin inhibitor based on the sanglifehrin scaffold NV651 as a candidate for HCC treatment. NV651 was shown to be a more potent cyclophilin inhibitor than CsA or SfA. Compared to sorafenib, NV651 presented a more potent antiproliferative effect in cancer cell growth in vitro and was able to decrease tumor growth in vivo, as well as induce cell cycle arrest in the G_2_/M phase specifically in the mitotic phase.

## 2. Materials and Methods

### 2.1. Cell Lines and Drugs

HEPG2 cells were purchased from Sigma-Aldrich (St. Louis, MO, USA) or American Type Culture Collection (ATCC) (Manassas, VA, USA). HUH7-D12 was purchased from Sigma-Aldrich (St. Louis, MO, USA). HEP3B and PLC/PRF/5 were purchased from ATCC (Manassas, VA, USA). HUH7 cells were from the Japanese Collection of Research Biosources Cell Bank (Osaka, Japan). LIXC-003, LIXC-004, LIXC-006, LIXC-012, LIXC-066 and LIXC-086 were established and characterized by Shanghai ChemPartner Ltd. (Shanghai, China), as previously described [29]. HEP3B2.1-7-LUC were provided by Shanghai ChemPartner Ltd. (Shanghai, China), and the parental HEP3B2.1-7 cells were purchased from ATCC (Manassas, VA, USA). HEPG2, HUH7, HEP3B2.1-7-LUC, LIXC-003, LIXC-004, LIXC-066, LIXC-012, LIXC-006 and LIXC-086 were authenticated by Short Tandem Repeat (STR) profiling. Experiments were performed between passages 2 or 3 up to passage 20 after thawing. HEPG2, HUH7-D12, HEP3B and PLC/PRF/5 were maintained in a high-glucose DMEM medium (Thermo Fisher Scientific, Cat# 11965-092, Waltham, MA, USA) supplemented with 10% fetal bovine serum (FBS) (Sigma Aldrich, Cat# F7524, St. Louis, MO, USA) and 1% penicillin/streptomycin (Thermo Fisher Scientific, Cat# 10378-016, Waltham, MA, USA). HUH7 cells were maintained in a low-glucose DMEM medium (Corning, Cat# 10-014-CMR, Corning, NY, USA) supplemented with 10% FBS and 1% penicillin/streptomycin. LIXC-003, LIXC-004 and LIXC-012 were maintained in RPMI1640 (Invitrogen, Cat# 11875-135, Waltham, MA, USA) supplemented with 10% FBS (Invitrogen, Cat# 10099-141, Waltham, MA, USA), 10-μg/mL insulin (Invitrogen, Cat# 12585-014, Waltham, MA, USA), 2-μM hydrocortisone (Sigma Aldrich, Cat# H0888-1G, St. Louis, MO, USA) and 2.5-mg/mL glucose. LIXC-086 was maintained with all the supplements indicated above, in addition to 10-ng/mL EGF (Invitrogen, Cat# PHG0311, Waltham, MA, USA). LIXC-066 was maintained with RPMI 1640, 10% FBS, 10-μg/mL insulin and 2-μM hydrocortisone and LIXC-006 in RPMI 1640 supplemented with 10% FBS, 10-ng/mL EGF, 10-μg/mL insulin and 2-μM hydrocortisone. HEP3B2.1-7-LUC was maintained with EMEM (Invitrogen Cat# 11095-080, Waltham, MA, USA) with 10% FBS. Cells were kept in a humified incubator at 37 °C in a 5% CO_2_ atmosphere. Cells were trypsinized with 0.25% trypsin (Corning, Cat# 25-053-CI, Corning, NY, USA). All cell lines were tested for Mycoplasma with the MycoAlert^TM^ Mycoplasma Detection Kit (Lonza, Cat# LT07-418, Basel, Switzerland) by PCR or by DAPI staining and analyzed by fluorescence microscopy. Sorafenib (Selleck Chemicals, Cat# S7397, Houston, TX, USA) was dissolved in DMSO at 10 mM, and NV651 was dissolved in DMSO at 0.5 or 10 mM and stored at −20 °C until use.

### 2.2. PPIase Activity Assay

The PPIase activity of CypA, CypB, CypD and Cyp40 was determined by following the rate of hydrolysis of N-succinyl-Ala-Ala-Pro-Phe-p-nitroanilide by chymotrypsin, a digestive enzyme that only hydrolyzes the trans form of the peptide. Therefore, the hydrolysis of the cis form, the concentration of which is maximized by using a stock dissolved in trifluoroethanol containing 470-mM LiCl, is only possible with the isomerization from the cis to the trans form. Cyps were equilibrated for 1 h at 5 °C with the selected concentrations of CsA, SfA or NV651. The reaction was started with the peptide addition, and the change in absorbance was monitored spectrophotometrically at a 6-s interval for 6 min. The blank rates of hydrolysis (in the absence of Cyps) were subtracted from the rates in the presence of Cyps. The initial rates of the enzymatic reaction were analyzed by a first-order regression analysis of the time course of the change in absorbance.

### 2.3. Toxicity Studies

#### 2.3.1. In Vitro

##### CellTiter Glo Studies

The evaluation of the off-target effects of NV651 was performed by ImQuest BioSciences Inc. (Frederick, MD, USA) by analyzing the cytotoxicity in a panel of primary cells. NV651 was evaluated at a concentration of 30 µM and five serial half-logarithmic dilutions in triplicate. The positive control compounds for each cell type were purchased from Sigma-Aldrich (St. Louis, MO, USA) and were as follows: 

Unstimulated human peripheral blood mononuclear cells (PBMCs), phytohemagglutinin (PHA)-stimulated PBMCs, monocytes/macrophages, dendritic cells, hepatocytes, induced pluripotent stem cell (iPS) neurons: Staurosporine, bone marrow progenitors: AZT, kidney cells (RPTEC): cisplatin and iPS cardiomyocytes: Doxazosin.

Evaluation of PBMCs

Human unstimulated or PHA-P-stimulated PBMCs were isolated by Ficoll hypaque centrifugation from whole blood and resuspended in a fresh tissue culture medium containing IL-2. Cells were seeded at a concentration of 5 × 10^5^ /mL in a 96-well plate followed by NV651 treatment in triplicate for 3 days. Cell viability was measured with CellTiter Glo (Promega, Madison, WI, USA).

Evaluation of Human Monocytes-Macrophages

Freshly separated pre-PHA-blasted PBMCs (from one donor) were suspended in DPBS with 10% heat-inactivated Human AB at 4 × 10^6^ cells/mL and seeded in a 96-well flat bottom microtiter plate. Plates were incubated for 2 h, followed by the incubation with RPMI1640 supplemented with 10% FBS, 2-mM L-glutamine, 100-U/mL penicillin and 100-µg/mL streptomycin and incubated for an additional 2 days. After this period, cells were washed and incubated for 5–7 days with the same media. Before the assay, the cell monolayer was washed once again to remove the residual PBMCs. Later on, complete RPMI media with the selected NV651 concentration was added in triplicate. Following a 7-day incubation, cells were stained with CellTiter Glo (Promega, Madison, WI, USA) to evaluate the cell viability.

Evaluation of Human Monocytes–Macrophages–Derived Dendritic Cells

The isolation was based on Nair, Archer [30]. Briefly, freshly separated PBMCs from one donor were suspended in DPBS at 4 × 10^6^ cells per mL and cultured in a T-75 flask. Cells were incubated for 30 min and then washed 5–7 times with DPBS to remove nonadherent cells. Cells were cultured in RPMI 1640 supplemented with 10% FBS, 2-mM L-glutamine, 25-mM HEPES, 100-U/L penicillin, 100-µg/mL streptomycin, 50-ng/mL rhGM-CSF and 50-ng/mL rhIL-4 and incubated for 7 days. After the incubation period, cells were cultured in the same media with the addition of LPS (10 ng/mL) for two days. Cells were washed once and resuspended at 1 × 10^6^/mL in the complete medium and plated in a 96-well plate, followed by addition of the compound in triplicate. After a 7-day incubation, CellTiter Glo (Promega, Madison, WI, USA) was used.

Evaluation of Primary Human Hepatocytes

Fresh primary human hepatocytes (BioreclamationIVT, Westbury, NY, USA) at a density of 7 × 10^4^ cells/well with a 0.2-mg/mL Matrigel overlay were received 24 h prior to the assay. The medium was then replaced with fresh hepatocyte culture medium (BioreclamationIVT, Westbury, NY, USA) and cultured overnight. The following day, fresh media was added containing the selected NV651 concentrations. The plate was cultured for 48 h before analysis with CellTiter Glo (Promega, Madison, WI, USA).

Evaluation of Human Bone Marrow Cells

Bone marrow progenitor cells (Invitrogen, Waltham, MA, USA) were suspended in cell culture medium (Iscove-modified Dulbecco medium) containing 15% FBS, 10% giant tumor-conditioned medium (Bone Marrow Plus, Sigma-Aldrich, St. Louis, MO, USA), 10-ng/mL recombinant human IL-6, 10-ng/mL recombinant human IL-3, 25-ng/mL human granulocyte macrophage colony stimulating factor (GM-CSF, R&D systems, Minneapolis, MN, USA) and a final concentration of 1% methylcellulose. Cells (1 × 10^5^ cells/well) were seeded in 6-well plates containing NV651 in triplicate. Cells were incubated for 14 days, and colonies (more than 30 cells) were counted.

Evaluation of iPS Cardiomyocytes

iPS cardiomyocytes (Cellular Dynamics, Madison, WI, USA) were seeded in a 96-well plate precoated with 0.1% gelatin (Sigma-Aldrich, St. Louis, MO, USA) in a plating medium (Cellular dynamics, Madison, WI, USA) at a concentration of 1.5 × 10^4^ cells/well. After 48 h of incubation, cells were washed, and NV651 diluted in the maintenance medium (Cellular Dynamics, Madison, WI, USA) was added. After 3 days of incubation, cell viability was measured with CellTiter Glo (Promega, Madison, WI, USA).

Evaluation of iPS Neurons

iPS neurons (Cellular Dynamics, Madison, WI, USA) were seeded at a concentration of 2 × 10^5^ cells/mL in 96-well plates coated with 0.01% poly-L-Lysine, 0.01% poly-L-ornithine (Cellular Dynamics, Madison, WI, USA) and 3.3-µg/mL of laminin solution. A specialized medium (Cellular Dynamics, Madison, WI, USA) was used for culturing. After 14 weeks of culturing, NV651 was added to the media in triplicate. Cells were treated for 3 days with NV651, followed by analysis with CellTiter Glo (Promega, Madison, WI, USA).

Evaluation of RPTEC Kidney Cells

Human primary renal proximal tubule cells (hRPTEC, Lonza, Basel, Switzerland) were cultured at 1 × 10^4^ cells/well in supplemented REGM medium (Lonza, Basel, Switzerland). Following overnight incubation for adherence, cells were treated with NV651 in triplicate. After a 3-day treatment, CellTiter Glo (Promega, Madison, WI, USA) was used.

CellTiter Glo

CellTiter Glo was performed to evaluate cell viability. CellTiter Glo Luminescent Cell Viability Assay (Promega, Madison, WI, USA) was performed according to the manufacturer’s instructions. Samples were measured with Wallac 1450 Microbeta Trilux liquid scintillation counter. Data was presented as IC_50_ (50% reduction in cell viability).

##### Primary Human Hepatocytes

This experiment was performed in Cyprotex (Watertown, MA, USA). Cryopreserved Primary human hepatocytes plated on 384 black-well plate cells were dosed with NV651 at a range of concentrations (0.195, 0.391, 0.781, 1.56, 3.13, 6.25, 12.5, 25, 50 and 100 µM). At the end of the incubation period (24 or 48 h), several health parameters were analyzed to evaluate the potential cytotoxicity of NV651 in primary human hepatocytes, including the total cell count with Hoechst staining and PI subtraction (a nonpermeable dye to quantify dead cells), nuclear intensity, Glutathione content (GSH), mitochondrial membrane potential, reactive oxygen species (ROS), steatosis, nuclear size and DNA structure. Plates were then scanned using an automated fluorescent cellular imager, ArrayScan (Thermo Scientific Cellomics).

Data was presented as the MEC: minimum effective concentration that significantly crosses the vehicle control threshold and the AC_50_ concentration in which a 50% maximum effect is observed for each cell health parameter.

#### 2.3.2. In Vivo

All the procedures related to animal handling, care and the treatment in this study were performed according to the guidelines approved by the Institutional Animal Care and Use Committee (IACUC) of Shanghai ChemPartner Ltd. (Shanghai, China) following the guidance of the Association for Assessment and Accreditation of Laboratory Animal Care (AAALAC). BALB/c nude mice were kept in individual ventilation cages at a constant temperature and humidity, with 5 animals in each cage. Animals had free access to irradiation-sterilized dry granule food during the entire study period and water. BALB/c nude mice between 6–8 weeks were treated with either 25, 50 or 200 mg/kg dissolved in 25% buffered beta OH cyclodextrin (*n* = 5 mice per treatment). Daily doses were given intraperitoneally (IP) for 5 days. Mice were weighted daily until the 6th day.

### 2.4. Pharmacokinetic Studies

These experiments were performed by Shanghai ChemPartner Co., Ltd. (Shanghai, China) following the guidance described above. 

#### 2.4.1. Single Oral Administration (PO)

Twelve CD1 male mice with a weight between 28–30 g were purchased from SLAC Laboratory Animal Co. Ltd. (Shanghai, China). NV651 was diluted in 5% ethanol, 5% Cremophor EL and 90% saline. A single 10-mg/kg dose of NV651 was given orally (PO). Blood and liver samples were taken at 1, 4, 8 and 24 h (*n* = 3). At the selected time points, 30 µL of the blood samples were taken via face glass insert or cardiac puncture and placed into K2EDTA tubes. Tissues were terminal-collected at the designated time points.

Thirty microliters of blood or liver samples were mixed with 100-µL Pierce Acetonitrile (ACN, Thermo Fisher Scientific, Waltham, MA, USA) containing 20-ng/mL IS (702) used as the internal standard. An aliquot of 10-µL supernatant was injected for LC-MS/MS analysis using LCMSMS-003 (API4000).

#### 2.4.2. Intravenous (IV) and Oral Administration (PO)

NV651 was prepared as described above. Six- to eight-week-old male mice were treated IV with 0.5 mg/kg or PO with 5 mg/kg, with 3 mice per group. Blood and plasma samples were taken pre-dose; at 5 min of treatment and at 1, 2, 4, 8 or 24 h. Analysis was performed as indicated above. 

### 2.5. Cell Proliferation Assay: Acumen

Compound screening and comparison of the antiproliferative effect of NV651 and sorafenib were performed by Shanghai ChemPartner Co., Ltd. (Shanghai, China). Cells were seeded in a 96-well plate at the following concentrations: LIXC-003 400 cells/well, LIXC-004 1000 cells/well, LIXC-086 5000 cells/well, LIXC-066 1250 cells/well, LIXC-012 2000 cells/well, LIXC-006 1250 cells/well, HEPG2 1000 cells/well, HUH7 2000 cells/well and Hep3B2.1-7-Luc 1000 cells/well. Cells were treated with either sorafenib or NV651 at 0.004, 0.011, 0.034, 0.103, 0.309, 0.926, 2.778, 8.333, 25 µM or an equivalent volume with DMSO.

For compound screening, HUH7 cells were treated at 0.0016, 0.008, 0.04, 0.2, 1 µM or an equivalent volume with DMSO.

After 7 days, cells were trypsinized and collected. Fifty microliters of cell suspension were added to 50 µL of 2-µM Calcein AM (Invitrogen, Cat# C3099, Waltham, MA, USA) diluted in HBSS and placed in a black-walled 96-well plate. The plate was left for 30 min at room temperature, followed by a scan with Acumen eX3.

Growth inhibition for direct cell counting (%) by Acumen was calculated as:(1)$$100-100*\frac{\mathrm{Experimental}\;\mathrm{count}-\mathrm{Blank}}{\mathrm{DMSO}\;\mathrm{count}\;-\mathrm{Blank}}$$

### 2.6. Cell Proliferation and Viability Assay with Acridine Orange and DAPI

For the quantification of the total cell number, cells were seeded at a concentration of 6000 cells/well for HEPG2 and 9000 cells/well for HUH7 in a 24-well plate. After 24 h of seeding, cells were treated with 10, 50 or 100 nM of NV651 or DMSO as the control dissolved in their correspondent media, fresh media with compound was added every 72 h. After 72 h of treatment, cells were harvested and stained with DAPI (Santa Cruz Biotechnology, Cat# SC3598, Dallas, TX, USA) and Acridine Orange (AO) (Sigma-Aldrich, Cat# A9231-10ML, St. Louis, MO, USA) at final concentrations of 5 µg/mL and 1.5 µg/mL, respectively. Quantification of the total cell number and nonviable cells was performed every day up to the tenth day. Total cell number was quantified with acridine orange staining, and the cell viability in each sample was quantified by the ratio between cells positive for DAPI staining (nonpermeable dye) and acridine orange staining.

### 2.7. Clonogenic Assay

Cells were seeded at 2500/mL and, starting 24 h later, provided with fresh medium and NV651 every day for the following 14 days. Then, colonies were stained with crystal violet, which was subsequently dissolved in 10% acetic acid and quantified by a spectrophotometer by absorbance at 595 nm.

### 2.8. Fluorocytometry

#### 2.8.1. Quantification of Mitochondrial Membrane Potential and Cell Permeability

HEPG2 and HUH7 cells were seeded at 1.5 × 10^5^ cells/well and 1 × 10^5^ cells/well, respectively, 24 h prior to treatment. After 72 h of treatment (0, 20, 50 or 100 nM of NV651), supernatant and trypsinized cells were collected and stained with propidium iodide (PI) (Thermo Fisher Scientific, Waltham, MA, USA) at a concentration of 1 μg/mL to evaluate the plasma membrane integrity; due to the lack of cell permeability, only dead cells are able to incorporate it. In addition, changes in the mitochondrial membrane potential were evaluated by 3,3′-dihexyloxacarbocyanine iodide (DiOC(6)3)(Molecular Probes–Invitrogen, Waltham, MA, USA) staining at a concentration of 40 nM.

#### 2.8.2. Cell Cycle Analysis

For the quantification of the DNA content, cells were treated with DMSO or 100 nM of NV651. Samples were collected at 24, 48, 72 and 96 h, followed by fixation with 70% (*v*/*v*) ethanol. DNA was stained with 50-µg/mL PI. Due to the capacity of PI to stain any type of nucleotide, RNase (100 µg/mL) (Sigma-Aldrich, Cat# R4875-100MG, St. Louis, MO, USA) was also added. To quantify the apoptotic cell fraction, we measured the subG_1_ fraction in the cell cycle.

#### 2.8.3. Phospho–Histone 3 (PH3) Staining

For the quantification of mitotic cells, HEPG2 cells were seeded in a 6-well plate at a concentration of 1.5 × 10^5^ cells/well and HUH7 at 1 × 10^5^ cells/well in a 6-well plate. Both cells lines were treated with 10 or 100 nM of NV651 or the equivalent volume of DMSO. Samples were fixed for HEPG2 at 24 h and for HUH7 at 48 h of treatment, as described above. Permeabilization was performed by adding 0.25% Tween20 (Sigma-Aldrich, Cat# P1379-500ML, St. Louis, MO, USA) diluted in PBS. Samples were stained with primary antibody mouse anti-PH3 (Merck Millipore, Cat# 05-806, Burlington, MA, USA) at a dilution of 1/100 overnight at 4 °C and with the secondary antibody goat anti-Mouse IgG AF488 conjugated (Life technologies, Cat# A11029, Carlsbad, CA, USA) for 1 h at room temperature. After washing, 1-µg/mL DAPI was added for the staining of the DNA content.

#### 2.8.4. Analysis

Data was acquired with FACSverse Instrument and analyzed with FCS Express 6 Flow Research.

### 2.9. Immunofluorescence

#### Mitotic Quantification

HUH7 cells were seeded at a concentration of 4.5 × 10^4^ cells/well in a 6-well plate containing coverslips, followed by the treatment of NV651 at 10 or 100 nM or the equivalent volume in DMSO. After 48 h, samples were fixed with 4% PFA (Histolab, Cat# 02176, Gothenburg, Sweden) and permeabilized with 0.1% Triton x-100 (Sigma-Aldrich, Cat# X100-500ML, St. Louis, MO, USA) diluted in PBS. Samples were stained with 1-µg/mL of rat anti-tubulin (Abcam, Cat# ab6161, Cambridge, UK) for one hour at room temperature. Lastly, DAPI and secondary antibody goat anti-rat IgG AF546 conjugate (Invitrogen, Cat# A11081, Waltham, MA, USA) were added. Coverslips were mounted on glass slides adding fluorescent mounting media (Dako, Agilent, Santa Clara, CA, USA). Thirteen pictures at 10× were taken with a Zeiss Confocal microscope and quantified for the number of mitotic cells in relation to the total number of cells with ImageJ.

### 2.10. In Vivo Efficacy

#### 2.10.1. Chemicals

Twenty-five percent buffered Hydroxy Propyl beta Cyclodextrin (HPbCD) was prepared from Hydroxy Propyl beta Cyclodextrin (Roquette, Lestrem, France), EDTA solution and phosphate buffer at a pH of 7.4, filtered with a 0.2-µM filter (VWR, Cat# 28145-483, Radnor, PA, USA) and stored at 4 °C. NV651 was prepared at a concentration of 10 mg/kg with 25% buffered HPbCD (vehicle). NV651 solution was sonicated for 30 min in a water bath sonicator (Bioruptor, Diagenode, Denville, NJ, USA) and shaken for up to an hour, followed by a secondary sterilization with a 0.2-µm filter. NV651 solution was stored at 4 °C up to 2 weeks, with stability confirmed up to 3 weeks.

#### 2.10.2. Cells and Mice

HEPG2 cells were transduced with RediFect Red-FLuc-GFP (Perkin Elmer, Cat# CLS960003, Waltham, MA, USA) with a MOI = 3 and sorted for GFP^+^ in a FACSAria III (BD Biosciences, Franklin Lakes, NJ, USA). GFP^+^Luciferase (Luc)^+^ HEPG2 cells were confirmed to be negative for mycoplasma with the MycoAlert^TM^ Mycoplasma Detection Kit (Lonza, Cat# LT07-418, Basel, Switzerland). Cell viability and a percentage of the GFP^+^ cells were analyzed before the start of the in vivo experiment with a BD FACSverse cytometer (BD Biosciences, Franklin Lakes, NJ, USA). Twenty NMRI nude female mice (6–8 weeks old) were injected in one flank with 1 million cells suspended in PBS with 50% Matrigel (Corning, Cat# 354230, Corning, NY, USA) in a total volume of 200 µL. All procedures were performed according to the national and international guidelines of the European Union and approved by the Swedish Regional (Malmö-Lund).

#### 2.10.3. Treatment and Analysis

Tumor growth was assessed with an in vivo imaging system (IVIS) one week after the injection of the cells. Once the tumor reached a sufficient bioluminescent signal (BLI) (total flux approx. 10^8^ photons/sec), mice with similar values were divided into either the vehicle or NV651 treatment group to have an equal starting value between groups (*n* = 10 mice per group). Both groups were treated daily for 3 weeks by subcutaneous injection with either 10 mg/kg of NV651 or an equal volume of vehicle (100 µL), and IVIS images were taken once a week. Body weights were measured at time 0 of the treatment and twice a week after the beginning of the treatment. Once the tumor was visible, caliper measurements were performed twice a week. Tumor growth was analyzed for up to 3 weeks. BLI signal intensity was quantified in total flux (photons/s) after deducting the average background signal from the measurement region of interest (ROI) around the tumor area using live image analysis software (PerkinElmer, Waltham, MA, USA).

### 2.11. Statistical Analysis

Statistical analysis was performed with GraphPad Prism 8.3.1. (San Diego, CA, USA). Data involving more than two groups and one time point were analyzed with one-way ANOVA, followed by Dunnett’s multiple comparison test. Data involving two or more groups at different time points were analyzed by 2-way ANOVA, followed by Sidak’s and Dunnett’s multiple comparison tests, respectively * *p* < 0.05, ** *p* < 0.01 and *** *p* < 0.001.

## 3. Results

### 3.1. Cyclophilin Overexpression in HCC

Initially, we investigated the relation between cyclophilin expression and HCC. Gene expression and Kaplan–Meier curves of PPIA; PPIB; PPIF and PPID (encoding for cyclophilin A, B, D and 40, respectively) were retrieved from Gene Expression Profiling Interactive Analysis (GEPIA) [31]. The expression of PPIA and PPIB was upregulated in the tumor in comparison to normal tissue and was correlated with a poor prognosis, with *p*-values of 0.0012 and 0.066, respectively, while PPIF and PPID did not show any significant differences (Figure 1).

### 3.2. NV651 Is More Potent Than Common Cyclophilin Inhibitors at Reducing PPIase Activity of Cyclophilins

NV651 is a member of a family of the optimized cyclophilin inhibitors based on the sanglifehrin scaffold (Figure 2A). The platform also includes NV556, which we previously have evaluated for the treatment of fibrosis [32]. NV651 was selected based on the potency of the antiproliferative activity (Appendix A). We then compared the inhibitory effect of NV651 with CsA and SfA on the PPIase activity in a free-cell assay. NV651 was shown to be more potent than the classical cyclophilin inhibitors CsA and SfA on CypA, CypB, CypD and Cyp40 (Figure 2B,C).

### 3.3. NV651 Showed Low Cytotoxic Activity against Normal Cell Types In Vitro and Promising In Vivo Tolerability

NV651 was evaluated in a panel of assays to determine the potential in vitro cytotoxicity to primary human cells, including monocytes–macrophages, dendritic cells, bone marrow progenitor cells, hepatocytes, human-induced pluripotent stem cell (iPS) cardiomyocytes, iPS neurons and primary renal proximal tubule epithelial cells (RPTEC). The NV651 cytotoxicity was evaluated with CellTiter Glo, and the colony formation assay generated IC_50_ values of 24.3, 24.2 and 20.9 µM against monocytes, dendritic cells and kidney cells, respectively. No other cell types displayed features of cytotoxicity at concentrations of NV651 up to 30 µM (Table 1A).

A detailed examination of the NV651 effect on primary human hepatocytes showed that the lowest minimum effective concentration (MEC) caused a decrease in the glutathione (GSH) content, suggesting the depletion of the GSH cellular pool at the µM range (Table 1B). The lowest AC_50_ response at 8.17-µM NV651 resulted in an increase in the ROS levels and an increase in toxic superoxide intermediates (Table 1B). Other cell parameters, including the nuclear intensity (increase in membrane permeability often linked to toxicity in hepatocytes), mitochondrial potential (mitochondrial toxicity, as well as apoptosis), nuclear size (DNA fragmentation), DNA structure (chromosomal instability and DNA fragmentation) and steatosis (accumulation of triglycerides within the cytoplasm of treated cells), provided evidence of the high therapeutic index of NV651, as no effects were observed until reaching the µM range in vitro. In addition, mice receiving a daily dose of 25 mg/kg, 50 mg/kg or 200 mg/kg of NV651 for 5 days via IP injection showed no body weight losses (Figure 3A).

We also assessed the pharmacokinetics (PK) of NV651 by two administration routes: intravenously (IV) or oral administration (PO) (Figure 3B). The blood levels in mice after a 0.5-mg/kg administration via IV or 5 mg/kg via PO NV651 were measured until the end of the experiment (24 h) (Figure 3B). This resulted in a peak in the concentration at about 5 min and 1 h using IV and PO administration, respectively, with an oral bioavailability of 2% (Figure 3B). This oral administration resulted in a T_max_ at 1 h in the blood and liver with NV651 and was still present 24 h after the treatment, with a three-fold higher level in the liver than blood (Figure 3C). These results showed that NV651 was orally bioavailable and accumulated in the liver.

### 3.4. NV651 Displays a More Potent Antiproliferative Effect Than Sorafenib in HCC Cell Lines

Since sorafenib is generally used as a first-line treatment for advanced HCC treatment, we compared the effects of NV651 with sorafenib in growth inhibition experiments using nine HCC cell lines (Figure 4A). A higher antiproliferative effect of NV651 in comparison to sorafenib in eight out of nine cell lines with an absolute IC_50_ up to 400 times lower could be observed (Figure 4B). A more detailed proliferation assay with the HEPG2 and HUH7 cell lines using different concentrations of NV651 for up to 10 days indicated an elevated doubling time from 43 to 827 h and from 35 to 155 h in HEPG2 and HUH7 cells, respectively (Figure 4C,D). To further corroborate these data, we performed clonogenic assays using four different HCC cell lines. The treatment of cells with NV651 reduced the number of colonies in all four cell lines (Figure 4E).

### 3.5. Prolonged Treatment of NV651 at a High Concentration Causes a Slight Increase in Cell Death

Given the ability of NV651 to inhibit cell proliferation, we decided to investigate whether NV651 promotes cell death by using DAPI staining, a nonpermeable cell dye, in relation to the total cell count. We could not observe any differences in the cell viability of HEPG2 but a significant decrease in HUH7 cells after 10 days of exposure (Figure 5A). Apoptotic cells were also analyzed by the % of the population with cleaved DNA (subG_1_ fraction) in HEPG2 and HUH7 cells treated with 100 nM of NV651 up to 96 h. This resulted in a statistically significant increase in cell death (Figure 5B). In contrast, an analysis of the changes in the mitochondrial membrane potential by DiOC6(3) staining, an early apoptotic marker, and the plasma membrane integrity by PI staining, a late apoptotic or necrotic marker, did not show any significant differences (Figure 5C,D). These data suggest that NV651 can induce minor cell death after a long treatment period, but the potent inhibition observed during proliferation could not be explained by cell death.

### 3.6. NV651 Induces Cell Cycle Perturbations

Next, we treated the HEPG2 and HUH7 cell lines with DMSO or 100 nM of NV651 for up to 96 h and stained the DNA content for the cell cycle studies. As shown in Figure 6A,B, a block in the G_2_/M phase was observed in both HEPG2 and HUH7 cells upon the NV651 treatment after 48–72 h. This accumulation was confirmed with the observation of an increased trend in the cell size in both HEPG2 and HUH7 cells (Figure 6C). In addition, the fraction of cells positive for the phosphorylation of histone 3 (PH3), a hallmark of mitosis, was increased in both HEPG2 and HUH7 cells after 24–48 of treatment with NV651 (Figure 6D,E). Confirming the previous results, an increase in the number of mitotic counts of HUH7 cells at 48 h upon NV651 exposure was also seen (Figure 6F).

### 3.7. NV651 Decreases Tumor Growth In Vivo

To evaluate the efficacy of NV651 in vivo, 20 female athymic nude mice were xenografted with Luc^+^ HEPG2 cells in the right flank. Tumor growth was assessed with the IVIS system once a week following randomization of the mice into the control (vehicle) and treatment (NV651) groups based on their initial BLI signals (total flux 1.5 × 10^8^ photons/s). When we observed a luminescence signal (1.5 × 10^8^ photons/s) with the in vivo imaging system, mice were treated daily by a subcutaneous injection with either the vehicle or 10 mg/kg of NV651. During the treatment period, no significant changes were observed in body weight (Figure 7B). After 3 weeks of treatment, the luminescence signal in the treatment group, representing tumor growth, was significantly reduced in comparison to the vehicle-treated group (Figure 7A,C). In addition, caliper measurements were performed twice a week once the subcutaneous tumor was visible. These measurements confirmed a reduced tumor growth in the NV651 treatment group (Figure 7D).

## 4. Discussion

Cyclophilins are emerging as interesting drug targets, as they are overexpressed in cancer [18,23,33]—specifically, in HCC [20,22,25]—and positively correlate with a worse prognosis. Cyclophilins control the correct folding of many proteins that are critical for malignancy [34]. In recent years, cyclophilins have also been shown to mediate the resistance to ionizing radiation [35] and cisplatin [25] and to support cancer growth [21,22,23] and metastasis [36]. Experiments performed with CsA, the most-known cyclophilin inhibitor, have shown a potential antiproliferative effect in HCC in the µM range [20]. On the other hand, CsA is well-known for its immunosuppressive activity due to the inhibition of the ternary complex formed by CypA with calcineurin [17].

In the present study, we characterized NV651, a novel SfA-based cyclophilin inhibitor demonstrated to be ten times more potent than SfA and a hundred times more potent than CsA, with no immunosuppressant activity [32]. Sorafenib is one of the main options for advanced HCC, prolonging the survival of patients up to three months [5]. In the growth inhibition experiments using nine different HCC cell lines, NV651 proved to be more potent than Sorafenib in eight out of nine, with IC_50_ values that could be achievable in vivo.

NV651 exerted its functions without presenting a potent cytotoxic effect. This is important, as the debris from cell death can recruit immune cells and sustain cancer growth [37]. Blocking proliferation without eliminating the cells can help maintain the tumor environment more stably and, therefore, more sensitively to other drugs [38,39]. NV651 was tested on human-derived normal cells where the IC_50_ values were above 20 µM, a much higher dose than the one active on HCC cells, indicating a good safety margin. 

The in vivo tolerability was tested in mice at three different doses with daily IP injections for 5 days, resulting in no significant weight loss or other clinical signs. We could also demonstrate that NV651 could be given via different administration routes such as intravenously or oral dosing, where the oral administration could be the safest and simplest option if NV651 is considered as a potential treatment for HCC. Due to the potent antiproliferative effect of NV651 and the low percentage of cell death, we investigated whether this antiproliferative effect is caused by a disturbance in the cell cycle profile. Previous studies have shown the importance of cyclophilins in the cell cycle. In gastric cancer and cholangiocarcinoma, cyclophilins have been reported to be involved in the G_1_/S transition [23,33]. The importance of cyclophilins in the G_1_/S transition has also been observed in HCC [20,21,22]. Alternatively, Jiang et al. [40] reported a G_2_/M arrest due to a CypA knockdown in lung adenocarcinoma cells. Indeed, the NV651 treatment caused an early accumulation of cells in the G_2_/M phase of the cell cycle. A more exhaustive analysis in this effect by quantifying the percentage of cells positive for the phosphorylation of H3 (PH3), a marker of mitosis, revealed an accumulation of cells in the mitotic phase.

The in vivo efficacy of NV651 was evaluated in xenograft nude mice after 3 weeks of treatment by the measurement of tumor growths with two independent methods: IVIS, a system able to analyze tumor growths at an early stage of tumor development, and by caliper measurement, where the evaluation of tumor volumes starts at a later stage of the tumor development. This last method was used to avoid any potential decrease of the bioluminescence due to necrotic tissue. Indeed, both methods of analysis confirmed the efficacy of NV651 to decrease tumor growth. In addition, no significant changes in body weight confirmed the safety of NV651 administration in vivo.

In this study, we did not evaluate the potential synergistic effect of NV651 in combination with the other treatments used against HCC. Previous publications have shown the ability of cyclophilins to facilitate therapy resistance in compounds such as cisplatin, a DNA-damaging reagent, by decreasing the ROS levels caused by this compound and modulating the DNA damage repair response [27,28]. In addition, cyclophilins have been linked to therapy resistance by reducing intracellular drug accumulation through the increased expression of ABC transporters [26]. Therefore, NV651, in combination with treatments against HCC, could present a synergistic effect.

## 5. Conclusions

In this study, we describe the development of NV651, a novel cyclophilin inhibitor for liver cancer treatment. NV651 showed potency to inhibit the proliferation of HCC cell lines via cell cycle perturbations and arrest of the cells in the mitotic phase. In addition, NV651 was able to decrease the tumor growth in a xenograft mouse model. We could also confirm the safety of NV651 in normal cells and good oral bioavailability, highlighting NV651 as a potential candidate for HCC treatment.

## 6. Patents

Gronberg et al. (2020). Use of sanglifehrin macrocyclic analogues as anticancer compounds (US 10,857,150 B2). United States Patent.

## Figures and Tables

**Figure 1 cancers-13-03041-f001:**
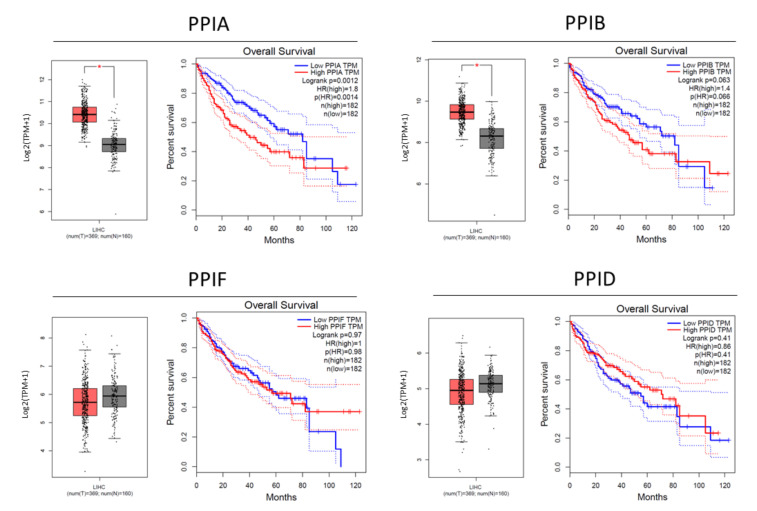
HCC gene expression and survival curves retrieved from GEPIA. Left graphs represent the gene expression levels of PPIA, PPIB, PPIF and PPID in HCC evaluated from The Cancer Genome Atlas (TCGA) tumors vs. TCGA normal + Genotype-Tissue Expression (GTEx) normal analyzed by one-way ANOVA. Tumor is indicated in red and normal tissue in grey. * *p* < 0.01 statistical difference in comparison to normal tissue. Right graphs represent Kaplan–Meier plots for the overall survival from 364 HCC patients in relation to the expression levels of the PPIA, PPIB, PPIF and PPID genes. Group cutoff is presented in a median at 50% for high and low.

**Figure 2 cancers-13-03041-f002:**
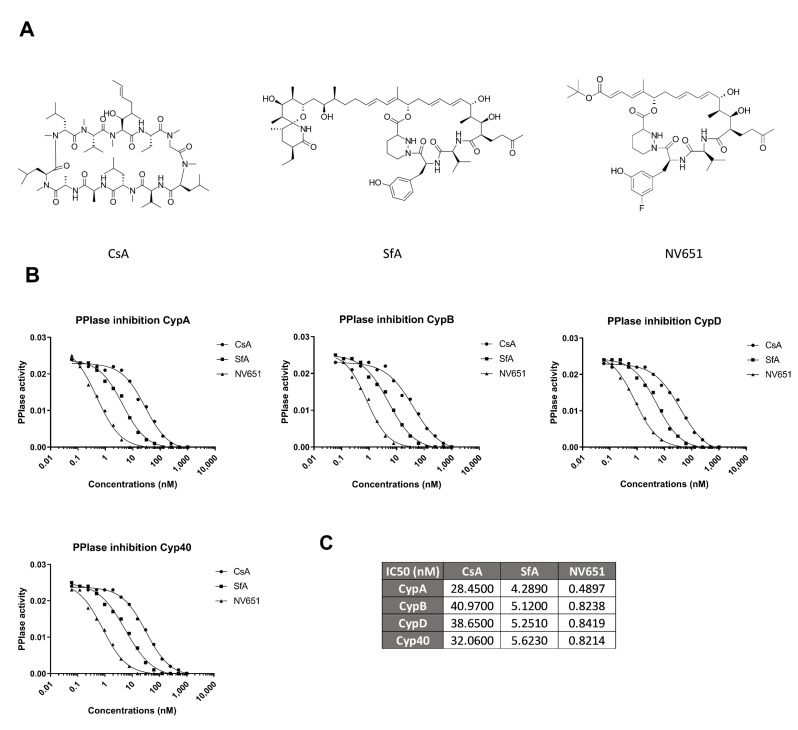
NV651 effect on cyclophilin activity. (**A**) Chemical structures of CsA, SfA and NV651. (**B**) PPIase activity of CypA, CypB, CypD and Cyp40 in the presence of different concentrations of the indicated drugs. The PPIase activity corresponds to the catalytic rate constants (seconds^−1^). (**C**) Collection presenting the absolute IC_50_ calculated from (**B**).

**Figure 3 cancers-13-03041-f003:**
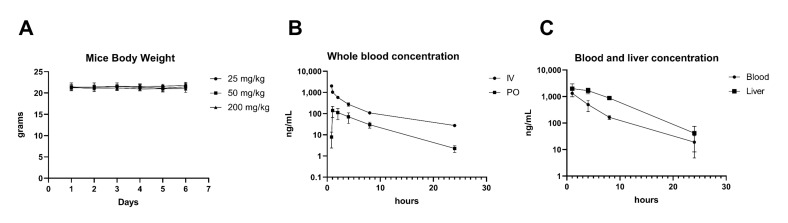
NV651 effect on body weight and PK analysis. (**A**) Mice body weight changes during the NV651 treatment in BALB/c nude mice, *n* = 5 mice per group. (**B**) NV651 concentration in the blood after an intravenous injection (0.5 mg/kg) or oral dosing (5 mg/kg) in CD1 mice, *n* = 3 mice per group. (**C**) NV651 concentration in the blood and liver after a 10-mg/kg administration PO in CD1 mice, *n* = 3 mice per group and time point.

**Figure 4 cancers-13-03041-f004:**
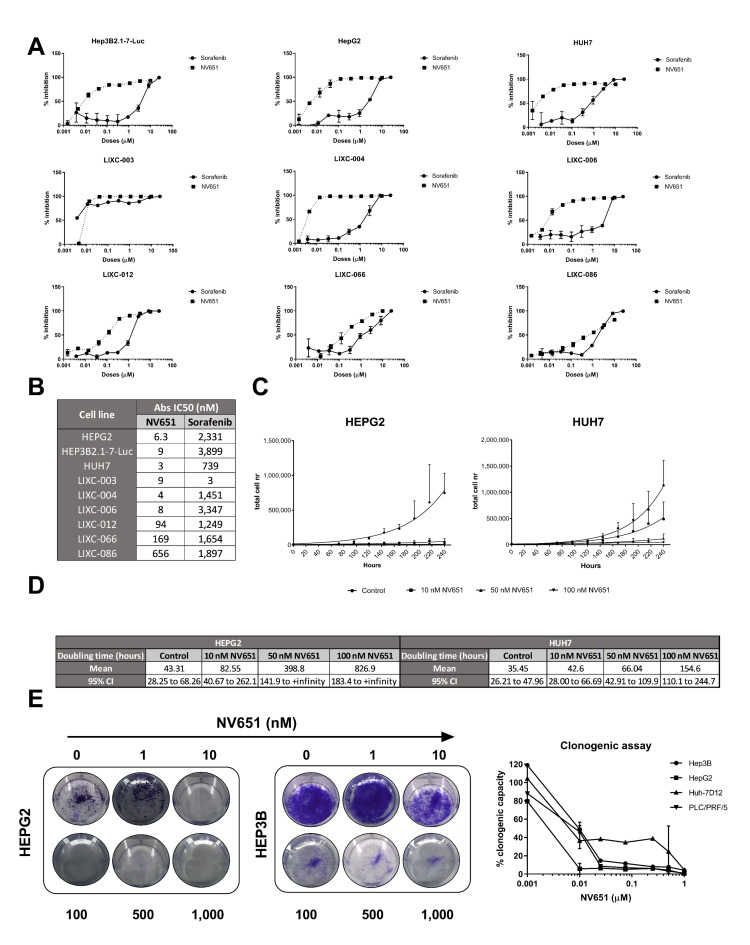
NV651 effect on cell proliferation. (**A**) Hep3B2.1-7-Luc, HepG2, HUH7, LIXC-003, LIXC-004, LIXC-006, LIXC-012, LIXC-066 and LIXC-086 were seeded in duplicate and treated for 7 days with the indicated drugs at different doses. (**B**) The table indicates the IC_50_ values for the different drugs in the different cell lines. (**C**) NV651 effect on HEPG2 and HUH7 cells for a treatment period of 10 days. The total cell number was quantified from 3 to 10 days and presented with cell growth curves, *n* = 3 to 4 biological replicates. (**D**) Doubling times: mean + 95% CI calculated from the growth curves in (**C**). (**E**) NV651 effect on the colony formation in Hep3B, HepG2, Huh-7D12 and PLC/PRF/5 seeded in triplicate and continuously treated at different doses for 14 days and stained with crystal violet. Data are presented as the mean ± SD.

**Figure 5 cancers-13-03041-f005:**
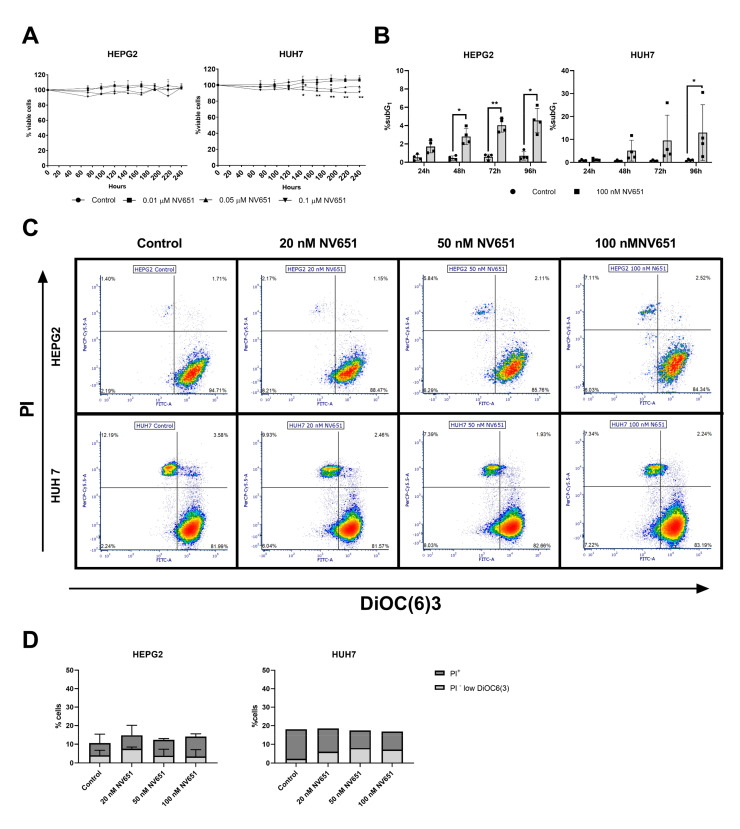
NV651 effect on cell death. (**A**) Nonviable cells are calculated by the ratio between DAPI^+^ cells and AO^+^ cells; % was normalized to day 0, and media with a new NV651 dose was added every 72 h during the experiment. (**B**) The subG_1_ fraction from Figure 6B after DMSO or 100-nM NV651 treatment for 24, 48, 72 or 96 h of treatment. (**C**,**D**) NV651 effect on the mitochondrial membrane potential (pre-apoptotic marker) and PI^+^ cells (late apoptotic + necrotic marker) after 72 h of exposure to the compound. (**C**) Representative density plot of HEPG2 and HUH7. (**D**) Quantification of PI^+^ cells and PI^−^ low DiOC(6)3, equivalent to the low mitochondrial membrane potential, (**A**,**B**) *n* = 3 to 4 biological replicates and (**D**) *n* = 1 to 2 biological replicates. (**A**) Statistically analyzed by 2-way ANOVA, followed by Dunnett’s multiple comparison test. (**B**) Statistically analyzed by 2-way ANOVA, followed by Sidak’s multiple comparison test. Data are presented as the mean ± SD. * *p* < 0.05 and ** *p* < 0.01.

**Figure 6 cancers-13-03041-f006:**
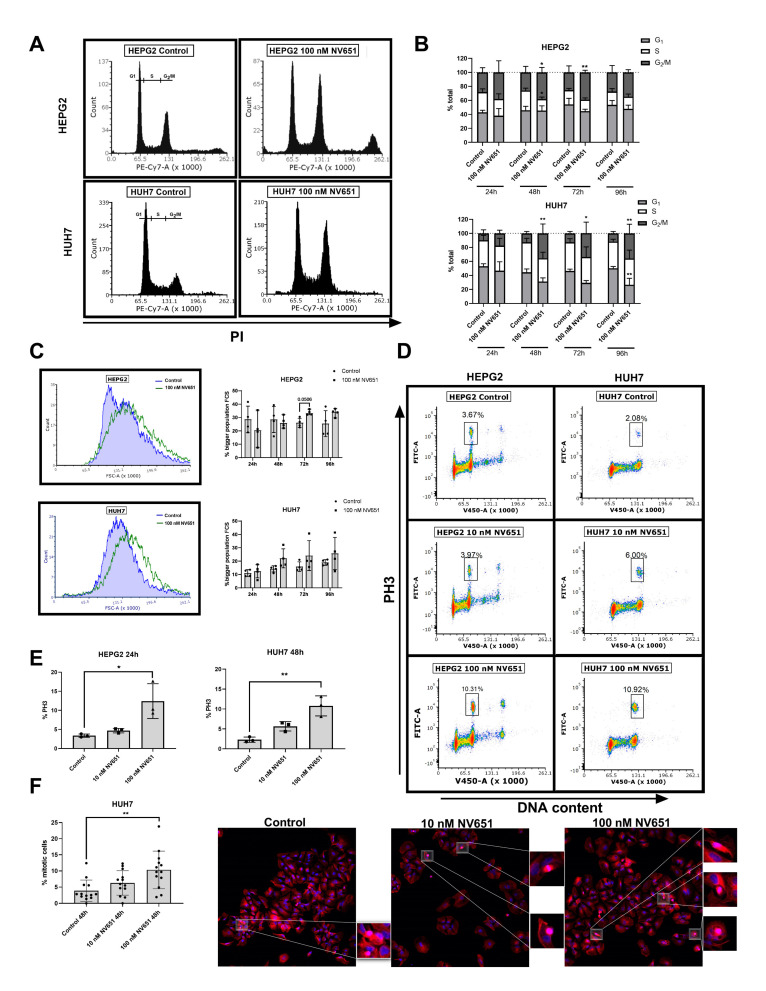
NV651 effect in the cell cycle. (**A**,**B**) DNA content analyzed with PI staining in HEPG2 and HUH7 treated with DMSO or 100 nM of NV651. (**A**) Representative histograms of HEPG2 at 24 h of exposure and HUH7 at 96 h. (**B**) G_1_, S and G_2_/M quantification at 24, 48, 72 and 96 h of NV651 treatment. (**C**) To the left, representative histograms of the FSC-A analysis of the samples in (**A**). To the right, quantification data at the selected time points. (**D**,**E**) PH3 staining of DMSO or 10 and 100-nM-treated HEPG2 or HUH7 cells at 24 and 48 h, respectively. (**D**) The representative density plots. (**E**) Quantification at specific time points. (**F**) Immunofluorescence staining with DAPI and tubulin in DMSO or 10- and 100-nM NV651 HUH7-treated cells at 48 h of treatment. To the left, the quantification of mitotic cells in 13 pictures at 10×. Data are presented as the mean ± SD. (**B**,**C**) Statistically analyzed by 2-way ANOVA, followed by Sidak’s multiple comparison test. (**E**,**F**) Statistically analyzed by one-way ANOVA, followed by Dunnett’s multiple comparison test. *n* = 4 biological replicates in (**B**,**C**), *n* = 3 biological replicates in (**E**) and *n* = 1 biological replicate in (**F**). Data are presented as the mean ± SD * *p* < 0.05 and ** *p* < 0.01.

**Figure 7 cancers-13-03041-f007:**
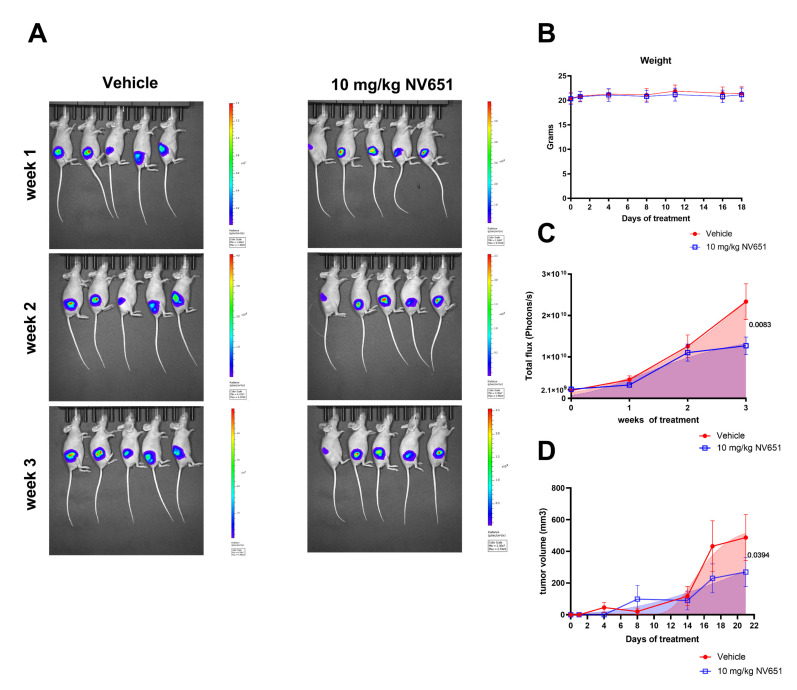
NV651 efficacy in nude mice xenografted with Luc+ HEPG2 cells after 3 weeks of treatment. (**A**,**C**) Luminescence intensity was measured once a week in the vehicle and NV651 groups for 3 weeks by the IVIS system. (**A**) Representative images of a luminescence signal in the vehicle or NV651-treated groups at 1, 2 and 3 weeks with a luminescence scale to their right. (**B**) Weight over time. (**D**) The subcutaneous tumor volumes were measured by caliper at the indicated time points. (**B**) Data were presented as the mean ± SD and statistically analyzed by 2-way ANOVA, followed by Sidak’s multiple comparison test. (**C**,**D**) Data were presented as the mean ± SEM and analyzed by a nonlinear regression analysis by Gomperth Growth with the *p*-value indicated, *n* = 10 mice per group.

**Table 1 cancers-13-03041-t001:** NV651 toxicity in normal cells. (A) IC_50_ values for NV651 and positive controls in various normal human cell lines. (B) NV651 MEC and AC_50_ values in primary human hepatocytes evaluated for several health parameters.

Table 1(**A**) IC_50_ values for NV651 and positive controls in various normal human cell lines			
	**Positive Control (µM) ^a^**	**NV651 (µM)**	**Days of Treatment**
Unstimulated PBMCs	0.01	>30	3
PHA-stimulated PBMCs	0.1	>30	3
Monocytes/Macrophages	0.02	24.3	7
Dendritic	0.009	24.2	7
Bone marrow progenitors	0.89	>30	14
Hepatocytes	1.02	>30	2
iPS cardiomyocytes	39.8	>30	3
iPS neurons	1.65	>30	3
RPTECs	0.009	20.9	3
Table 1(**B**) NV651 MEC and AC_50_ values in primary human hepatocytes evaluated for several health parameters			
**Cell health Parameter (Hepatocytes)**	**Time (h)**	**MEC (µM)**	**AC50 (µM)**
Nuclear intensity	24	8.90	8.94
Mitochondrial Membrane potential	24	12.40	14.20
Glutathione content	24	2.62	10.90
ROS	24	6.43	8.17
Cell count (PI corrected)	48	6.14	13.90
Nuclear size	48	18.10	21.50
DNA structure	48	10.90	34.38
Steatosis	48	6.81	39.00

^a^ The positive control compounds for each cell type were as follows: PBMCs, PHA-stimulated PBMCs, monocytes/macrophages, dendritic cells, hepatocytes, iPS neurons: Staurosporine, bone marrow progenitors: AZT, RPTEC: cisplatin and iPS cardiomyocytes: doxazosin.

## Data Availability

Gene expression and survival curves were retrieved from GEPIA http://gepia.cancer-pku.cn/ (accessed on 23 April 2021).

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
