# Peer review of "Novel Cyclophilin Inhibitor Decreases Cell Proliferation and Tumor Growth in Models of Hepatocellular Carcinoma"

_cancers, 2021, doi:10.3390/cancers13123041_

Round 1
Reviewer 1 Report
The paper by Serrano et al. describes the characterization of a small molecule, NV651, as an inhibitor of peptidyl prolyl isomerase A and B and as an potential anti-tumor agent for HCC. The investigation is thorough, the data convincing, and the manuscript well prepared. In my view, this manuscript is acceptable for publication after some minor revisions.
Here are a few minor concerns that should be addressed before publication.
1) In the Figure 1 survival graphs, the Y-axis is labeled as "Percent survival", but the values are not percentages, but fractions of 1. This should be corrected.
2) In Fig 2B, the PPIase activities have no units. It should be explained what the activities actually are.
3) Table 1 compared the toxicity of NV651 to a positive control, but I could not find any information on the positive control. This information should be given in the legend to the table and in Materials and Methods.
The subtitle of 3.7 is "NV651 decreases cell growth in vivo". What was actually reported is "tumor growth in vivo". This should be revised.
Author Response
Reply to reviewer:
Response: We thank the Reviewer for the careful review of our manuscript and the valuable and constructive suggestions.
Here are a few minor concerns that should be addressed before publication.
1) In the Figure 1 survival graphs, the Y-axis is labeled as "Percent survival", but the values are not percentages, but fractions of 1. This should be corrected.
Response: The graphs were directly retrieved from GEPIA http://gepia.cancer-pku.cn/
2) In Fig 2B, the PPIase activities have no units. It should be explained what the
activities actually are.
Response: We have included the units for the PPIase activity in the figure legend.
3) Table 1 compared the toxicity of NV651 to a positive control, but I could not find any information on the positive control. This information should be given in the legend to the table and in Materials and Methods.
Response: We have included information about the positive control in the figure
legends (Table) and Material & Methods.
The subtitle of 3.7 is "NV651 decreases cell growth in vivo". What was actually
reported is "tumor growth in vivo". This should be revised.
Response: We have replaced "NV651 decreases cell growth in vivo” with "tumor
growth in vivo".
Reviewer 2 Report
Currently available treatment options for hepatocellular carcinoma (HCC) are based on tyrosine kinase inhibitors. The results in clinical practice are unsatisfactory because of important adverse reactions and only a few of advanced HCC cases with complete and/or partial responses. Thus, there is an unmet need for novel effective treatments. The work of Sonia Simón Serrano and co-workers is very interesting and their results are promising for new perspectives of treatment The in vitro and in vivo evidences illustrate a potential efficacy of the novel cyclophilin inhibitor NV651 which significantly decreased the tumor proliferation in different HCC cell lines causing the accumulation of cells in the G2/M phase mitosis and consequently reducing tumor growth in vivo in the HCC-xenograft model. The lack of body weight loss in the animals underlines the very promising safety profile of NV651 observed in the toxicity study in primary human hepatocytes. Thus the manuscript provide a well-studied and convincing evidence of the therapeutic potential of the novel NV651 cyclophilin inhibitor.
Author Response
We thank the Reviewer for the careful review of our manuscript and highlighting the importance of our finding.